# Screening for Diabetes Mellitus in Patients with Hidradenitis Suppurativa—A Monocentric Study in Germany

**DOI:** 10.3390/ijms24076596

**Published:** 2023-04-01

**Authors:** Nessr Abu Rached, Thilo Gambichler, Lennart Ocker, Johannes W. Dietrich, Daniel R. Quast, Christina Sieger, Caroline Seifert, Christina Scheel, Falk G. Bechara

**Affiliations:** 1International Centre for Hidradenitis Suppurativa/Acne Inversa, Department of Dermatology, Venereology and Allergology, Ruhr-University Bochum, 44791 Bochum, Germany; 2Diabetes, Endocrinology and Metabolism Section, Department of Internal Medicine I, St. Josef Hospital, Ruhr University Bochum, NRW, Gudrunstr. 56, 44791 Bochum, Germany; 3Diabetes Centre Bochum-Hattingen, St. Elisabeth-Hospital Blankenstein, Im Vogelsang 5-11, 45527 Hattingen, Germany; 4Centre for Rare Endocrine Diseases, Ruhr Centre for Rare Diseases (CeSER), Ruhr University Bochum and Witten/Herdecke University, Alexandrinenstr. 5, 44791 Bochum, Germany; 5Centre for Diabetes Technology, Catholic Hospitals Bochum, Gudrunstr. 56, 44791 Bochum, Germany

**Keywords:** hidradenitis suppurativa, HS, acne inversa, hormones, diabetes mellitus, metabolic disorder, insulin resistance, insulin, glucose metabolism disorder, screening, prevention, psoriasis, obesity

## Abstract

Hidradenitis suppurativa (HS) is a chronic skin disease that is often associated with metabolic disorders. Diabetes mellitus (DM) is a frequent comorbidity in HS. There is currently no established screening for DM in HS patients. The aim of our study was to identify high-risk groups of HS patients that develop DM and to assess the frequency of different types of DM present in HS patients. To do so, we conducted a monocentric study in 99 patients with HS. All patients underwent detailed clinical and laboratory assessments, including the determination of glycated hemoglobin. Among the 20.2% of patients that presented with DM, type 2 was by far the most prevalent (19 out of 20 patients). Moreover, male gender, age, BMI, Hurley stage, modified Hidradenitis Suppurativa Score (mHSS), DLQI and hypertension all correlated with the glycated hemoglobin levels in the HS patients. In the multivariable analysis, Hurley stage III, older age, and higher BMI were significantly associated with DM. Specifically, patients at Hurley stage III were at a 5.3-fold increased risk of having DM type II compared to patients at earlier Hurley stages. Since many of the HS patients had not been diagnosed, our study reveals shortcomings in the screening for DM and suggest that this should be routinely performed in HS patients at high risk to avoid secondary complications.

## 1. Introduction

Hidradenitis suppurativa (HS) is a chronic inflammatory skin disease that is commonly associated with metabolic syndrome [1]. In addition, HS is associated with a reduced quality of life of affected patients, which completely impedes normal functioning [2]. Current therapy for HS includes a variety of local therapies, antiseptic measures, antibiotics, biologics, surgery, and new-small molecule drugs [3,4]. HS is also associated with endocrine abnormalities and comorbidities [5]. Among these, diabetes mellitus (DM) may be the most common endocrine comorbidity in HS patients. In two BMI-adjusted meta-analyses, the pooled odds ratios for DM were 2.17-fold (95% CI, 1.9–2.6) and 2.8-fold (95% CI, 1.8–4.3) higher in HS patients compared to controls [6,7]. The US and Canadian Hidradenitis Suppurativa Foundations already recommend screening for DM in HS patients by determining fasting blood glucose or glycated hemoglobin, but do not identify high-risk groups of HS patients who are at an increased risk for DM [8]. By contrast, the current German guidelines for the management of HS do not contain any recommendations regarding DM. In fact, in Germany, no standardized guidelines with respect to screening for DM exist at all. Here, we wished to address this gap in the early detection of DM, and thereby the prevention of secondary diseases, specifically in HS patients. Indeed, some studies suggest that the role of DM and insulin resistance in HS is severely underestimated, which is most likely at least in part due to a lack of standardized screening for DM in HS patients [9]. Importantly, DM contributes to a wealth of secondary diseases, such as a high risk of cardiovascular disease, and impaired quality of life, all of which are simultaneously addressed upon treatment of chronically perturbed glucose metabolism. As a corollary, the treatment of diabetes mellitus appears to also improve symptoms of HS, at least in some patients [10]. For example, a phase 3 study is currently investigating whether metformin leads to an improvement in HS symptoms, disease severity and insulin sensitivity (ClinicalTrials.gov NCT04649502). The aim of the present study was to investigate the prevalence of different types of DM, and the impact on the quality of life in HS patients as a basis for recommendations concerning DM screening in routine clinical practice. Moreover, a particular focus was placed on identifying HS patients with a high risk for developing or presenting with DM.

## 2. Results

### 2.1. Personal and Diabetes Mellitus-Specific Characteristics

All personal and DM-specific characteristics are shown in Table 1. Of a total of 99 HS patients, 46.5% (n = 46) were female and 53.5% (n = 53) male. The mean age of the patients was 41.6 ± SD 13 years and the median of HS onset was 20 years (IQR 17-30). About one third of the patients had a positive family history of HS. A large proportion of the HS patients were active smokers (n = 63, 64.9%). The average BMI was 31.5 kg/m^2^ ± SD 6.7. About 14% of the patients were currently receiving treatment with the TNF-α inhibitor adalimumab. In 20.2% of the patients (n = 20), diabetes was detected. Men with HS were more likely to have DM than women with HS. Thus, 70% of the HS patients with diabetes were male and 30% female. In the male HS group, the proportion of patients with diabetes was 26.4% and in females, the proportion was 13%. In four patients, screening revealed DM that was previously unknown. The most common antidiabetic therapy in our HS cohort was metformin (n = 8, 40%). Three patients presented with insulin-dependent DM and two patients did not require antidiabetic medication. We also analyzed the prevalence of different types of diabetes mellitus in HS. Most of the HS patients with DM had type 2 diabetes (n = 19, 95%). In one case, type 1 DM was present. Diabetes mellitus type 3 or 4 were not detected. The most common DM cluster type in HS was mild obesity-related diabetes (MOD) at 75%, followed by cluster type severe insulin-deficient diabetes (SIDD) at 15%. In our screening, another 23 patients had prediabetes.

### 2.2. Differences between HS Patients with and without DM

To investigate the differences between HS patients with and without DM, we performed the non-parametric Mann–Whitney U test (Table 2). The mean age of the patients with DM was 49.7 ± 11 years and in the HS group without DM, it was 39.6 ± 12.7 years. The mean age differed significantly and was higher in HS patients with DM (*p* = 0.001). However, there was no difference concerning the duration of HS and first manifestation of the disease (*p* = 0.3 and 0.16, respectively). HS patients with DM had a significantly higher BMI than patients without DM (34.8 ± 6.1 kg/m^2^ vs. 30.6 ± 6.6 kg/m^2^; *p* = 0.008). HS disease severity differed significantly between the two groups, with respect to the Hurley stage and mHSS. In patients with DM, the prevalence of Hurley III was 70% and in patients without DM, the prevalence of Hurley III was only 39.2% (*p* = 0.01). In addition, the modified scoring system Hidradenitis Suppurativa Score (mHSS) also revealed a difference between the two groups (*p* = 0.03). Quality of life assessed by DLQI, and subjective pain showed no difference (*p* = 0.09 and 0.26, respectively). The DM group showed no difference with respect to positive family history for HS and smoking status compared to the non-DM group (*p* = 0.22 and 0.74, respectively). Other comorbidities characteristic for HS, such as hypothyroidism, polycystic ovary syndrome (PCOS), hypertension, psoriasis, and acne vulgaris/conglobata, did not differ significantly between the two groups (all *p* > 0.05).

Not surprisingly, the levels of glycated hemoglobin (HbA1c) correlated significantly with age and BMI (Table 3; r = 0.43, *p* < 0.001 and r = 0.42 *p* < 0.001; respectively). However, the degree of HS, as determined by the Hurley classification, also correlated with HbA1c levels (r = 0.3 and *p* = 0.002), albeit comparatively weakly. A similar weak correlation was observed between HbA1c levels and mHSS, the latter representing the number of inflammatory lesions (r = 0.28 and *p* = 0.005).

### 2.3. Identification of HS Patients with a High Risk for DM

To determine whether the disease parameters of HS patients were predictive for the presence of DM, we formulated a logistic regression model with DM as the dependent variable in order to calculate the odds ratios (OR, Table 4 and Figure 1). For this purpose, we included all variables from the univariable analyses with a *p* value ≤ 0.05. Thus, we determined that age and BMI showed an increased OR of 1.1 (95% confidence interval (CI) 1.03–1.16; *p* = 0.005) and 1.2 (95% CI 1.02–1.25; *p* = 0.019), respectively. Interestingly, compared to other Hurley stages, patients at Hurley III stage were 5.3-fold more likely to present with DM (95% CI 1.01–27.9; *p* = 0.048). The remaining parameters, male gender, mHSS, DLQI and hypertension, were not significant.

## 3. Discussion

In this study, DM was shown to be a common comorbidity of HS, mainly occurring in patients with severe HS, as defined by Hurley stage III. In our HS cohort, the prevalence of DM was 20.2%, which is comparable to previously published data [12]. Here, we specifically distinguish between different types of DM and report a majority of type 2 diabetes in HS patients. Since chronic inflammation is driving both DM and HS, it is likely that there is significant overlap with respect to the degree of molecular characteristics of the inflammatory response and severity of both DM and HS. For example, chronic inflammation in type 2 DM is associated with increased levels of pro-inflammatory macrophages and the recruitment of B-cells and T-cells that promote insulin resistance [13]. Activation of macrophages via inflammatory cytokines then causes pancreatic beta cell dysfunction [13]. Loss of β-cell function leads to the development of type 2 DM. The inflammatory process that operates via the IL-1 β pathway is one of the most well-studied pathways in type 2 DM [14], as well as in HS [15]. Given this overlap in inflammatory response between HS and DM, it is interesting to consider whether commonly employed anti-inflammatory therapy for HS, for example with TNF-α inhibitors, also leads to an improvement of the hyperglycemic metabolic situation. However, whether TNF-α inhibitors improve hyperglycemia and DM is controversially discussed [16,17,18,19,20,21,22,23]. Nonetheless, an overall agreement exists that TNF-α inhibitors have an impact on insulin sensitivity. In our study, there were no significant differences between glycated hemoglobin levels and TNF-α inhibitor therapy with adalimumab. However, the patient group on adalimumab in our study was too small to assess this conclusively (14 patients). Thus, on the one hand, further prospective studies are needed to determine whether anti-inflammatory therapy in HS patients influences insulin sensitivity and disease severity. On the other hand, many case series and reports show improvement in HS disease severity in patients on antidiabetic medication [5,10]. This further suggests that the disease severity of DM and HS may be, at least partially, linked through common mechanisms and negatively impact each other, presumably by promoting chronic inflammation in a vicious, proinflammatory circle. The proportion of DM-cluster-type MOD is very high in HS compared to other German cohorts (75% vs. 32%) [24]. Herder et al. were able to show that the different DM cluster types differed in the inflammatory pathway and level of inflammatory markers [24]. The results suggest that inflammation plays an important role in the development of DM in HS. Furthermore, our results highlight the importance of adipose tissue as an endocrine organ in the evolution of HS. Some studies in HS showed a disbalance of adipokines compared to the control subjects, independently of BMI [5]. We propose that adipose tissue in HS should be further investigated regarding its function and structure.

A link between chronic inflammatory dermatoses and DM has also been noted in other contexts. For example, hyperactivation of mechanistic target of rapamycin complex 1 (mTORC1) signalling is thought to be a driver of DM type 2 in HS [25]. Upregulation of mTORC1 has also been reported in other inflammatory dermatoses such as psoriasis and may explain the commonly observed development of insulin resistance in psoriasis patients [26].

Older age and a higher BMI were both correlated with a higher risk for DM, as expected with an OR of 1.1 (95% CI 1.03–1.16) for age and 1.2 (95% 1.02–1.25) for BMI [27,28]. From the results, it can be interpreted that high BMI and older age also have an influence on the development of DM. The likelihood of developing insulin resistance increases with age and BMI. It can be concluded that a high BMI is also a cause of DM. Interestingly, the age of HS patients with DM compared to HS patients without DM was about 10 years higher (49.7 ± 11 vs. 39.6 ± 12.7; *p* = 0.001). In our view, DM screening in HS is recommended from the age of 38 at the latest.

A prospective Danish cohort study investigated the prevalence of antidiabetic drugs being prescribed for patients with HS [29]. Importantly, there was no significant difference in the prevalence of antidiabetic drugs between the HS patients and non-HS patients, even though HS patients are significantly more likely to have DM than non-HS patients. Together, these data suggest that in many HS patients, DM has not been correctly diagnosed. This fact could be because screening measures for DM are not yet standardized and need to be improved. This is particularly pertinent for groups with an increased risk for diabetes, as exemplified by HS patients. In our cohort, in 18 HS patients with DM, antidiabetic therapy was necessary due to the markedly increased levels of HbA1c. However, of these 18 patients, 22% had not yet received any therapy because DM had not been diagnosed. Thus, our data underscore that improvement and standardization of screening recommendations is urgently needed. Although the German obesity guideline recommends DM screening for obese people, our study shows that screening needs to be improved in HS patients [30]. Many HS patients do not take advantage of screening measures because patients are usually younger and rarely visit a non-dermatologist. However, HS patients are more likely to visit a dermatologist because of skin lesions and inflammation. A recommendation regarding this issue in the HS guideline could improve screening.

In our study, the prevalence of DM in HS patients differed significantly between Hurley stages (*p* = 0.01). Among the patients at Hurley stage I, no patient had DM, at Hurley II, 13% of the patients had DM and at Hurley III, 31.1% of patients had DM. In conclusion, the prevalence of DM in HS patients at Hurley stage III was more than twice as high compared to Hurley stage II. Similar results were also presented in a hospital-based cohort study by Jørgensen et al. [31]. They reported a prevalence of 2.6% at Hurley stage I, 8.4% at Hurley stage II and 24.4% at Hurley stage III [31]. In addition, we calculated that HS patients at Hurley stage III were 5.3-fold more likely to have DM than HS patients with Hurley I or II (95% CI 1.01–27.9; *p* = 0.048). Indeed, a significant relationship between HS disease severity and DM has been described before [32]. However, to our knowledge, our study is the first work that aimed to define a high-risk group among HS patients for developing DM. Given that the overall comorbidity burden, and hence mortality rate, is higher in HS than psoriasis patients, the identification of high-risk groups, such as patients with Hurley stage III for diabetes, and subsequent optimization of screening procedures appears particularly important [33].

German guidelines recommend DM screening by HbA1c or fasting blood glucose in older people, as these methods are more minimally invasive than an oral glucose tolerance test (OGTT) [34]. For this reason, we did not perform OGTTs for uniformity. The omission of the OGTT is a limitation of our study. Performing an OGTT could further increase DM prevalence in HS. Fasting blood glucose levels were below 126 mg/dl in all patients with HbA1c values between 5.7% and 6.4%.

Compared to register studies, our cohort of 99 patients is comparatively small. Due to the limited sample size, some negative results (e.g., missing association with PCOS, hypertension or hypothyroidism) may be confounded by the low statistical power. Revisitation in larger studies may help to address these remaining questions. Nevertheless, our data are comparable to the known data from the literature. However, a strength of our study is the strong clinical correlation. Thus, our data yield directly translatable recommendations for screening, indicating that HS patients at Hurley stage III should be frequently screened for DM starting at a relatively young age, a recommendation that should be considered for inclusion in the forthcoming guidelines on the management of HS.

## 4. Materials and Methods

In this monocentric investigation between February 2022 and January 2023, data from 99 HS patients were collected at the International Centre for Hidradenitis suppurativa/Acne inversa Bochum. Diagnosis of HS was independently confirmed by two experienced dermatologists. In cases of discrepancies, another experienced dermatologist was consulted. All patients without complete data were excluded from this study.

All patients underwent DM screening by the examination of HbA1c levels and fasting blood glucose levels, as well as by the collection of the history of known comorbidities. The diagnosis of DM was made according to the current German guidelines [35]. DM was present with an HbA1c > 6.5% and an HBA1c between 5.7% and 6.4% was considered prediabetes. A fasting blood glucose level ≥ 126 mg/dL was also an indicator of DM. HS patients with DM were subdivided according to the clusters of the All New Diabetics In Scania (ANDIS) study [11,36]. The cluster classification was based on the presence of antibodies, age at diagnosis of DM, BMI and HbA1c [11]. Diabetes types 1–4 were established based on medical history, known comorbidities, presence of antibodies and initial manifestation.

Data are reported as the absolute number and percentage, median and interquartile range (IQR) or mean ± standard deviation (±SD) in all sections. We performed a sample size calculation using two-sample comparison (confidence level 95%; significance level 0.05; power 0.8). The first group of the two-sample comparison was the German DM prevalence of 7.2% and the second group was the expected DM prevalence at HS of 20%. We calculated that at least 88 patients were needed for our study. The Shapiro–Wilk test and Q-Q-Plot were used to test for normal distribution. The variables age and BMI were normally distributed and the remaining variables were not normally distributed. To investigate differences between the HS patients with and without DM, we performed the Mann–Whitney U test in Section 2.2. To detect significant relationships between the variables, we performed the Spearman correlation test in Section 2.2. Finally, we used a logistic regression model to exclude potential confounders and identify multi-variable relationships between the variables in Section 2.3. All variables with a *p* value < 0.05 in the univariable analyses were included in the logistic regression. The statistical analysis of all the data was carried out with IBM SPSS Statistics (version 29.0.0.0). A *p*-value < 0.05 was considered as significant.

## 5. Conclusions

In summary, DM was confirmed to present an important comorbidity in HS patients that should also be considered in the treatment of HS. Regular screening of HbA1c or fasting blood glucose levels should be performed in patients from the 4th decade of life with severe forms of HS, especially Hurley stage III. A recommendation to screen for DM in this high-risk group should be included in the forthcoming guidelines on clinical management of HS. Patients with Hurley III have a 5.3-fold increased risk of developing DM compared to Hurley I and II patients. A high BMI was also associated with an increased risk of DM.

## Figures and Tables

**Figure 1 ijms-24-06596-f001:**
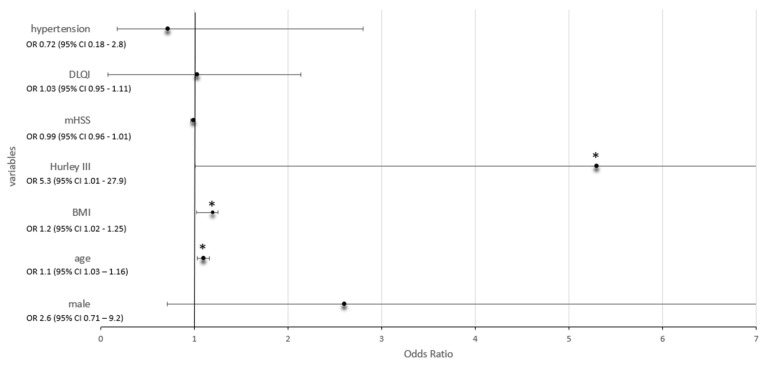
Odds ratio (OR) plot showing the different ORs in relation to different variables (hypertension, DLQI, mHSS, Hurley III, BMI, age, male); BMI, body mass index; mHSS, modified Hidradenitis Suppurativa Score; DLQI, Dermatology Life Quality Index; * significant result.

**Table 1 ijms-24-06596-t001:** Personal and diabetes mellitus-specific characteristics of HS patients.

Parameter		Value (s)
Sex, n (%)	Female	46 (46.5)
	Male	53 (53.5)
Age, mean (± SD), y		41.6 (±13)
Age of HS onset, median (IQR), y		20 (17–30)
Disease duration, median (IQR), y		13 (7–25)
BMI, mean (± SD), kg/m^2^		31.5 (±6.7)
Family history of HS, n (%)	Positive	31 (31.3)
	Negative	68 (78.7)
Smoker, n (%)	Current smoker	63 (64.9)
	Ex-smokers	3 (3.1)
	Non-smoker	31 (32)
Therapy with TNF-α inhibitor, n (%)	Current therapy	14 (14.1)
	Never	77 (77.8)
	Discontinued therapy	8 (8.1)
Prediabetes, n (%)	Total	23 (23.2)
Diabetes mellitus, n (%)	Total	20 (20.2)
	Male	14 (70)
	Female	6 (30)
Fasting blood glucose level, median (IQR), mg/dL	No diabetes mellitus	92 (86–96)
	Prediabetes	100 (88–116)
	Diabetes mellitus	137 (83–182)
Glycated hemoglobin, median (IQR), %	No diabetes mellitus	5.2 (5–5.4)
	Prediabetes	5.8 (5.7–6)
	Diabetes mellitus	7.8 (6.9–8.9)
Prevalence of DM using Hurley classification, n (%)	Hurley I	0 (0)
	Hurley II	6 (13)
	Hurley III	14 (31.1)
Type of DM, n (%)	Type 1 DM	1 (5)
	Type 2 DM	19 (95)
	Type 3 DM	0 (0)
	Type 4 DM	0 (0)
Diabetes cluster according to ANDIS [11]	Cluster 1 SAID	0 (0)
	Cluster 1 LADA	1 (5)
	Cluster 2 SIDD	3 (15)
	Cluster 3 MOD	15 (75)
	Cluster 4 MARD	0 (0)
	Cluster 5 SIRD	0 (0)
Insulin-dependent DM, n (%)		3 (15)
Antidiabetic medication, n (%)	No previous therapy	4 (20)
	No therapy necessary	2 (10)
	Insulin therapy	3 (15)
	Biguanides (metformin)	8 (40)
	GLP-1 receptor agonists	3 (15)
	SGLT-2 inhibitors	2 (10)
	DPP-4 inhibitor	1 (5)
Initial diagnosis of DM, n (of all in %/DM patients in %)		4 (4/20)

n, absolute number of patients; SD, standard deviation; y, years; IQR, interquartile range; BMI, body mass index; HS, hidradenitis suppurativa; DM, diabetes mellitus; ANDIS, All New Diabetics in Scania; SAID, severe autoimmune diabetes; LADA, latent autoimmune diabetes of adults; SIDD, severe insulin-deficient diabetes; MOD, mild obesity-related diabetes; MARD, mild age-related diabetes; SIRD, severe insulin-resistant diabetes; GLP-1, glucagon-like peptide; SGLT-2, sodium glucose linked transporter 2; DPP-4, dipeptidyl peptidase-4.

**Table 2 ijms-24-06596-t002:** Presentation of differences between HS patients with and without diabetes mellitus using non-parametric Mann–Whitney U test (n = 99).

Parameters	All Patients	HS Patients without DM	HS Patients with DM	*p* Value
Male vs. female, n (%)	53 (53.5) vs. 46 (46.5)	39 (49.4) vs. 40 (50.6)	14 (70) vs. 6 (30)	0.1
Age, mean (±SD), y	41.6 (±13)	39.6 (±12.7)	49.7 (±11)	**0.001 ****
Age of onset, median (IQR), y	20 (17–30)	20 (16–28)	23.5 (18–46.8)	0.16
Disease duration, median (IQR), y	13 (7–25)	13 (7–24)	19 (6.5–29.8)	0.3
BMI, mean (±SD), kg/m^2^	31.5 (±6.7)	30.6 (±6.6)	34.8 (±6.1)	**0.008** ******
Positive family history of HS, n (%)	31 (31.3)	27 (34.2)	4 (20)	0.22
Current smoker, n (%)	63 (63.6)	50 (63.3)	13 (65)	0.74
Hurley III, n (%)	45 (45.5)	31 (39.2)	14 (70)	**0.01 ***
mHSS, median (IQR)	40 (21–72)	33 (21–64)	64 (32.8–99.3)	**0.03 ***
DLQI, median (IQR)	13 (6.8–19.3)	12 (5.8–18.3)	16 (11.3–21.5)	0.09
Number of episodes in last 4 weeks, median (IQR)	1 (0–3)	0 (0–3)	1.5 (0–3)	0.18
Pain during visit (NRS), median (IQR)	2 (0–6)	2 (0–5)	4 (0–7.8)	0.26
Glycated hemoglobin, median, IQR, %	5.5 (5.1–5.9)	5.4 (5.1–5.7)	7.6 (6.5–8.4)	**<0.001 *****
Hypertension, n (%)	33 (33.3)	23 (29.1)	10 (50)	0.08
Hypothyroidism, n (%)	18 (18.2)	12 (15.2)	6 (30)	0.13
Psoriasis, n (%)	4 (4)	2 (2)	2 (10)	0.14
Acne vulgaris/conglobata, n (%)	15 (15.2)	13 (16.5)	2 (10)	0.47
PCOS, n (%)	3 (3)	3 (3.8)	0 (0)	0.38
Current adalimumab, n (%)	14 (14.1)	11 (13.9)	3 (15)	0.9

BMI, body mass index; HS, hidradenitis suppurativa; DM, diabetes mellitus; n, absolute number of patients; SD, standard deviation; y, years; IQR, interquartile range; NRS, numeric rating scale; mHSS, modified Hidradenitis Suppurativa Score; DLQI, Dermatology Life Quality Index; PCOS, ppolycystic ovary syndrome; * significant result; ** very significant; *** highly significant.

**Table 3 ijms-24-06596-t003:** Correlation analysis by Spearman rank coefficient between HbA1c levels and factors (n = 99).

Parameters	r	*p* Value
Male	0.31	0.002 **
Age	0.43	<0.001 ***
Disease duration	0.13	0.2
BMI	0.42	<0.001 ***
Positive family history of HS	−0.003	>0.9
Current smoker	0.09	0.4
Hurley III	0.3	0.002 **
mHSS	0.28	0.005 **
DLQI	0.22	0.03 *
Hypertension	0.22	0.03 *
Hypothyroidism	−0.04	0.7
Psoriasis	0.13	0.2
Acne vulgaris/conglobata	−0.06	0.6
PCOS	−0.11	0.3
Current adalimumab	−0.14	0.17

BMI, body mass index; modified Hidradenitis Suppurativa Score; HS, hidradenitis suppurativa; DLQI, Dermatology Life Quality Index; PCOS, ppolycystic ovary syndrome; r, Spearman rank coefficient; * significant result; ** very significant; *** highly significant.

**Table 4 ijms-24-06596-t004:** Logistic regression with the dependent variable of diabetes mellitus and independent variables from the univariable analyses with a *p* value ≤ 0.05 (n = 99).

Parameters	Odds Ratio (OR)	95% Confidence Interval (CI)	*p* Value
Male	2.6	0.71–9.2	0.2
Age	1.1	1.03–1.16	0.005 **
BMI	1.2	1.02–1.25	0.019 *
Hurley III	5.3	1.01–27.9	0.048 *
mHSS	0.99	0.96–1.01	0.22
DLQI	1.03	0.95–1.11	0.47
Hypertension	0.72	0.18–2.8	0.6

BMI, body mass index; mHSS, modified Hidradenitis Suppurativa Score; DLQI, Dermatology Life Quality Index; * significant result; ** very significant.

## Data Availability

The data presented in this study are available upon request from the corresponding author. The data are not publicly available due to privacy restrictions.

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
