# Peer review of "Screening for Diabetes Mellitus in Patients with Hidradenitis Suppurativa—A Monocentric Study in Germany"

_ijms, 2023, doi:10.3390/ijms24076596_

Round 1
Reviewer 1 Report
Many thanks for the opportunity to read this paper.
The interplay of hormonal factors and factors such as obesity, DM, insulin resistance is crucial in HS patients.
The paper is interesting, noteworthy, written in good sounding scientific language. I have minor comments :
1) abbreviations - under figures and tables , are not described everywhere
2) in material and methods section there is lack of information about statistical analysis, however tests for particular parts of results are described but it needs to be systematized e.g. subsection statistical analysis in M'n'M section. With a detailed description of the tests and the software on which these statistical tests were carried out
3) The Introduction lacks important information about the nature of the disease and its treatment i.e. no sentence about the reduced quality of life in these patients that completely disables normal functioning e.g. PMID : 35893421
There is a missing sentence about the treatment of HS itself : Treatment with antibiotics and new biologic drugs, small molecules drugs etc. Also missing is information that HS is treated surgically PMID : 36686007 and that there are recent methods of surgical treatment of HS like co-graft ADM and STSG , PMID : 36004913 and the very use of ADM in surgery and surgical treatment of HS e.g. PMID : 36359387 and the important work of Chafin et al. PMID : 33175626.
4) the discussion is interestingly conducted,- a small digression : I would add a piece about the research conducted about surgical treatment of obesity and the effect on insulin resistance, diabetes and HS itself. There are such studies being conducted.
5) to assess severity you used only Hurley? Hurley is a very good scale I am very happy that you only used Hurley. However, knowing the readers mainly dermatologists, ISH4 would be better seen. I do not require you to redo the work and use ISH4 but it is worth mentioning that there is something like this for severity assessment.
Congratulations on your interesting paper. It is very good and with the suggested changes it will be excellent.
With best regards,
Author Response
The answers can be found in the appendix.

Reviewer 2 Report
The manuscript entitled „Screening for diabetes mellitus in patients with hidradenitis suppurativa - a cross-sectional study in Germany” presents interesting issue, but some problems should be corrected.
The manuscript presented seems to be just an idea for the further study, but not a scientific research, due to a serious methodological problems.
Authors define their study as a “prospective cross-sectional monocentric study”, but it seems that they are not familiar with the definitions, as in fact they did not conduct the prospective study, but just presented a characteristics of a group of patients from 1 hospital. Authors should be aware that:
(1) The prospective study watches for outcomes, such as the development of a disease, during the study period and relates this to other factors such as suspected risk or protection factor(s). Taking this into account, they should study a group of patients with hidradenitis suppurativa (preferably newly diagnosed), without diabetes mellitus and observe this population for ages in order to observe development of the diabetes mellitus.
(2) The prospective study usually involves taking a cohort of subjects and watching them over a long period. Taking this into account, the assessment of less than 100 participants does not meet the criteria. Similarly, observation conducted even for 1 year does not, but it must be emphasized, that (based on the information presented in the manuscript) it seems that Authors studied each participant only once during the study period, so it was not observation, but single assessment.
Taking this into account, the conducted study was not a “prospective cross-sectional monocentric study”, as declared by Authors. At the same time, they may intended to present a simple screening, but for such study they need much larger sample (even larger than for prospective study). The sample size should be properly calculated for it and the calculation should be conducted based on a proper data.
Author Response

(The authors gave the same response as above.)

Reviewer 3 Report
The authors make the readers aware of the underdiagnosed comorbidity (DM) in patients with HS.
I have a comment on the authors not mentioning any other risk factors for diabetes 2 in their patients.
The higher risk for DM compared to controls in 2 meta-analyses are mentioned: Were the controls healthy individuals, i.e., no obesity or hypertension, or only not having HS?
Obesity and insulin resistance are risk factors for DM. To really show risks contributable to HS the control group should have similar BMI but have less DM. Please mention any such study.
Is it the HS or any obesity that is assumed to heighten the risk for DM? Put another way - the German guidelines do not mention any routine testing for DM in HS patients, but are there guidelines that prompt testing for dm in obese persons? Would HS patients with no obesity have the same risk for DM as those with obesity and HS?
Would patients with BMI of almost 35 (The DM patients in your study) not have been screened for DM even if they didn't have HS? You do show that there is significant difference according to obesity. How can you assume that it is the HS and not obesity that is the cause of DM? Would obese patients not be screened for diabetes if they did not have HS? Or will it be solely the HS that prompts screening.
Did you collect data on family history for diabetes? You mention family history on HS. Shouldn't also diabetes family history be mentioned? Would obese patients with a family history for diabetes not be screened for DM if they didn't have HS or family history of HS?
I suggest these issues be thoroughly discussed.
How did you use BMI in the logistic regression? Groups? < or than...>? continuous? Other?
Please could you also discuss the element of duration. Could it be expected that more advanced Hurley stage is seen when longer duration, and could this mean longer duration of obesity?
The same where age is concerned. Will it be expected that older patients would have had HS longer and also be more prone to DM because of age?
In the discussion, would you consider also discussing the role of fatty tissue as an endocrine organ and the role it may have in chronic inflammation and developing insulin resistance?
Please comment if also obesity improves on antidiabetic medication when you mention that antidiabetic medication improves HS. Can it be the lower BMI that leads to lower HS severity?
Typo: 2.1. Personal and diabetes mellitus specific characteristics characteristics
RRemove the extra 'characteristics'
Author Response

(The authors gave the same response as above.)

Round 2
Reviewer 1 Report
Authors well adressed all suggestions.
accept in current form
Author Response
Thank you for the review.
Reviewer 2 Report
The manuscript entitled „Screening for diabetes mellitus in patients with hidradenitis suppurativa - a monocentric study in Germany” presents interesting issue, but some problems should be corrected.
The manuscript presented seems to be just an idea for the further study, but not a scientific research, due to a serious methodological problems.
In the previous version of the manuscript, Authors claimed that they prepared a “prospective cross-sectional study”, but after my review, they changed it into just “screening”. But it does not solve all the existing problems:
(1) The screening requires much larger sample (even larger than for prospective study). The sample size should be properly calculated for it and the calculation should be conducted based on a proper data. I indicated it within my previous review, but Authors did not present the calculation
(2) The screening can not be conducted in one hospital only, as does not present “screening”, but just a hospital report
(3) Screening must be prepared based on rigorous criteria, while Authors did not. They declared that they diagnosed diabetes mellitus based on the German criteria (https://link.springer.com/article/10.1007/s11428-021-00763-7), based on HbA1c values, but the referred criteria does not allow to diagnose diabetes mellitus based on HbA1c values only, as for values of HbA1c of 5.7-6.5% the values of 2h OGTT must be taken into account (for values of HbA1c of 5.7-6.5%, only for 2h OGTT of > 200 mg/dl the diabetes mellitus may be diagnosed)
(4) Authors did not indicate how did they verify the participants of the study for type of diabetes and did not even present obtained values of HbA1c (which based on their Materials and methods Section were the most important variable for screening)
Author Response
See in the attachment.

Reviewer 3 Report
The authors have given point-by-point answers to my comments, but not provided point by pint changes in the paper. Changes that are made to the paper are almost not possible to assess since whole discussion and introduction are deleted and rewritten where the individual changes are not visible. A reviewer cannot compare sentence-by-sentence/word-by-word the previous text with the updated text.
I still do not see any mention of BMI potentially being the reason for DM, rather than the HS, and did not see a discussion on this.
Please also make clear to the reader that the 2 meta-analyses were adjusted for BMI.
If German guidelines say that persons with high BMI should be screened for DM and HS patients, who also are obese, have not been screened it should be pointed out that obviously these guidelines are not being followed. Will guidelines on HS screening for DM be expected to work better?
The discussion and conclusion should include those points. If these actually are mentioned subtly somewhere in the text this should be clear for the reviewer by pointing out the specific changes.
Author Response
see in the attached appendix

Round 3
Reviewer 2 Report
The manuscript entitled „Screening for diabetes mellitus in patients with hidradenitis suppurativa - a monocentric study in Germany” presents interesting issue, but some problems should be corrected.
The manuscript presented seems to be just an idea for the further study, but not a scientific research, due to a serious methodological problems.
In the previous version of the manuscript, Authors claimed that they prepared a “prospective cross-sectional study”, but after my review, they changed it into just “screening”. But it does not solve all the existing problems:
(1) The screening requires much larger sample (even larger than for prospective study). The sample size should be properly calculated for it and the calculation should be conducted based on a proper data (Authors should present within their manuscript estimated sample size, confidence level, margin of error and estimated proportion of population). I indicated it within my previous review, but Authors did not present the calculation within their manuscript.
(2) The screening can not be conducted in one hospital only, as does not present “screening”, but just a hospital report
(3) Screening must be prepared based on rigorous criteria, while Authors did not. They previously declared that they conducted it based on HbA1c values, but while I indicated that the referred criteria does not allow to diagnose diabetes mellitus based on HbA1c values only, as for values of HbA1c of 5.7-6.5% the values of 2h OGTT must be taken into account (for values of HbA1c of 5.7-6.5%, only for 2h OGTT of > 200 mg/dl the diabetes mellitus may be diagnosed), now they declare that they used the fasting glucose also. However, it also does not solve the problem, as still they need to have also 2h OGTT.
(4) Authors did not indicate how did they verify the participants of the study for type of diabetes
Author Response

(The authors gave the same response as above.)

Reviewer 3 Report
The authors have made adequate changes to the paper.
I only have a comment on this sentence: 'In German guidelines, DM screening by HbA1c or fasting blood glucose is recommended 246 in older people because they have few side effects' What do you mean by side effects? Do you mean is little invasive?
Author Response
"The authors have made adequate changes to the paper.
I only have a comment on this sentence: 'In German guidelines, DM screening by HbA1c or fasting blood glucose is recommended 246 in older people because they have few side effects' What do you mean by side effects? Do you mean is little invasive?"
Answer: We thank you for the good correction suggestions. Our point was that the tests (HbA1c or fasting blood glucose) are more minimally invasive than the OGTT and that is why screnning by HbA1c and/or fasting blood glucose is recommended in the current German guideline for older people. Screening by OGTT is not recommended in older people. We have changed the corresponding passage to "minimally invasive".
Round 4
Reviewer 2 Report
I have nothing more to add to Authors
Author Response
Thank you for your review.